# CausalFairness: An Open Source Python Library for Causal Fairness Analaysis

## Abstract

As machine learning (ML) systems are increasingly deployed in high-stakes domains, the need for robust methods to assess fairness has become more critical. While statistical fairness metrics are widely used due to their simplicity, they are limited in their ability to explain why disparities occur, as they rely on associative relationships in the data. In contrast, causal fairness metrics aim to uncover the underlying data-generating mechanisms that lead to observed disparities, enabling a deeper understanding of the influence of sensitive attributes and their proxies. Despite their promise, causal fairness metrics have seen limited adoption due to their technical and computational complexity. To address this gap, we present CausalFairness, the first open-source Python package designed to compute a diverse set of causal fairness metrics at both the group and individual levels. The metrics implemented are broadly applicable across classification and regression tasks (with easy extensions for intersectional analysis) and were selected for their significance in the fairness literature. We also demonstrate how standard statistical fairness metrics can be decomposed into their causal components, providing a complementary view of fairness grounded in causal reasoning.

## 1 Introduction

Statistical fairness metrics are easy to compute but only capture associations (i.e. conditional probabilities), not causality — limiting their ability to identify whether observed statistical disparities are truly caused by protected attributes or not. Causal fairness metrics, based on Structural Causal Models (SCMs) [6], overcome this by attributing observed disparities to specific sources (protected attributes, mediators or confounders). They also enable causal decompositions of statistical fairness metrics, thus delivering deeper insights into fairness audits. Despite their value, causal metrics are rarely used in practice due to technical challenges: they are harder to compute, require do-interventions, and face identifiability constraints. Each causal fairness metric often needs a custom architecture, and disagreement over the causal graph adds complexity. To address this, we introduce **CausalFairness**—the first open-source Python package to implement generalizable algorithms for key, established causal fairness metrics in the literature, including *Counterfactual Effects* [7]/[12], *Counterfactual Equalized Odds* [11], and *Counterfactual Fairness* [3] (see Table 1 and Appendix A.1 for an overview). The package is designed to work with minimal identifiability assumptions[1], does not (necessarily) require a fully specified SCM and supports both group and individual-level fairness metrics. We demonstrate CausalFairness on three datasets—*Adult Income*, *COMPAS*, and

---

[1]This is not equivalent to us making the claim that we address the challenge of identifiability or choice of cause of graph. Instead, the paper focuses on implementing metrics for which identifiability constrains are not very strong and thus can be implemented for a wide variety of problems and domains without running into problems of identifiability; For causal model discovery, see packages like `CausalNex`, `DoWhy`, or `CausalML`

Submitted to 39th Conference on Neural Information Processing Systems (NeurIPS 2025). Do not distribute.

33 *LSAC*—and provide code for replication.[2]. The paper positions itself as being a novel contribution to
34 the causal fairness literature by implementing novel computational algorithms for computing three
35 existing causal fairness metrics in the CausalFairness package, thus filling a critical gap in enabling
36 practical use of causal fairness metrics[3]. This work contributes to the counterfactual measurement
37 branch of causal fairness literature [14] (see Appendix A.2 for a brief literature review).

## 2   Methodology

39 **Notation and Preliminaries** : Causal fairness is typically formalized using Structural Causal Models
40 (SCMs)[6], where a Directed Acyclic Graph (DAG) represents observed variables (nodes) and their
41 causal relationships (edges). $Y$ is the true outcome, $\hat{Y}$ the predicted outcome, and $y$ the favorable
42 outcome (e.g., $y = 1$ in the Adult Income dataset). $A$ is the set of protected attributes (e.g., race,
43 gender), $X$ contains all features excluding $A$, and $a_0, a_1$ denote advantaged and disadvantaged groups.
44 The causal effect of an intervention $do(X = x)$ is expressed via the counterfactual distribution
45 $P(Y_x = y)$, where $Y_x$ is the outcome had $X$ been set to $x$. Often, $P(y|x) = P(Y = y|X = x)$ is
46 used interchangeably. The *Standard Fairness Model*[12] (see Appendix A.3) outlines three causal
47 pathways from $A$ to $Y$ and $\hat{Y}$: **Direct Path** $(A \rightarrow Y)$: Captures direct discrimination (e.g., gender or
48 race directly influencing income or recidivism), interpreted as *disparate treatment*; **Indirect Path**
49 $(A \rightarrow M \rightarrow Y)$: Effects mediated by variables like education or prior offenses, indicating *disparate
50 impact*; **Spurious Path** $(A \leftarrow C \rightarrow Y)$: Non-causal associations due to confounders $(C)$, such as
51 country of residence or age/gender, also contributing to *disparate impact*. Discrimination is assessed
52 on the DAG using counterfactuals. By applying different $do$-interventions, specific paths can be
53 isolated. The key idea is *ceteris paribus*, how does changing the protected attribute $A$ affect $Y$ or $\hat{Y}$?

### 2.1   CausalFairness : Definitions and Computational Algorithms

55 The core of the *CausalFairness* package is the `CausalFairnessDecomposition` class (see Ta-
56 ble 1), built on the standard fairness model [12]. It accepts $X, Y, \hat{Y}$, lists for $A, M$, optionally $C$,
57 derived from an SCM (algorithmically discovered or expert-curated) or a DAG, and a task-type flag
58 (regression/classification). The class provides three main methods: `analyse_mean_difference`
59 – group-level analysis using the Counterfactual Effects framework; `analyse_equalized_odds`
60 – causal decomposition of error rates; `analyse_counterfactual_fairness` – individual-level
61 fairness analysis. Each method compares the outcome (acceptance rate, error rates, or predicted
62 outcome $\hat{Y}_i$) in two counterfactual worlds: one with observed $A$, and one with counterfactual $A$. The
63 first two rely on (estimated) conditional probabilities using Gaussian Mixture Models for scalability;
64 the third requires a DAG and fits a graphical causal model.

Table 1: Pseudo-Algorithm for Causal Fairness Metrics

| A. Counterfactual Effects (Mean Difference) | B. Counterfactual Equalized Odds (EO) | C. Counterfactual Fairness |
|---|---|---|
| **Inputs:** $D, A, M, C, a_0, a_1, y$ | **Inputs:** $D, A, C, a_0, a_1, y, \hat{f}$ | **Inputs:** $A, M, C, a_0, a_1$, DAG |
| **1.** For each $(m, c) \in D$: 
 - Compute: $\mathbb{E}(Y = y \mid a_0, m, c)$ 
 - Compute: $\mathbb{E}(Y = y \mid a_1, m, c)$ 
 **2.** Estimate via GMM: 
 $\quad P(m \mid a_0, c), P(m \mid a_1, c)$ 
 $\quad P(c \mid a_0), P(c \mid a_1)$ 
 **3.** Combine expectations and probabilities 
 $\quad$ to compute the counterfactual effects | **1.** For each $c_j \in D$: 
 - Predict: $\hat{f}(c_j, a_0), \hat{f}(c_j, a_1)$ 
 - Obtain: $P(\hat{y}_{a_0, c_j}), P(\hat{y}_{a_1, c_j})$ 
 **2.** Estimate via GMM: 
 $\quad P(c \mid a_0), P(c \mid a_1)$ 
 **3.** Combine predictions and probabilities 
 $\quad$ to compute the Cft-EO | **1.** Fit SCM using DAG and dataset $D$ 
 **2.** For each individual $i \in D$: 
 - Get $A_{obs}$ (observed) and $A_{cf}$ (counterfactual) 
 - Sample from SCM under: 
 $\quad do(A = A_{obs}) \Rightarrow D_{obs}$ 
 $\quad do(A = A_{cf}) \Rightarrow D_{cf}$ 
 - Predict: $\hat{f}(D_{obs}), \hat{f}(D_{cf})$ 
 - Check: $Y_{obs} \neq Y_{cf}$ |

### 2.2   Counterfactual Effects : How does the protected attribute affect the predicted outcome?
###        Calculating Disparate Treatment, Disparate Impact and Explaining the Causal
###        Mechanism Behind Observed Statistical Parity

68 Counterfactual effects [13] is a family of three causal measures of discrimination related to statistical
69 parity, namely: **Counterfactual Direct Effect (Ctf-DE)**: measures direct discrimination along

---

[2]Code will be released; cleared on 10/9/25

[3]The implementation has internationally been optimized so as to not require any special hardware requirements

$A \rightarrow \hat{Y}$ by holding $M$ and $C$ constant, isolating the effect of $A$ on $\hat{Y}$. [7] define symmetric Ctf-DE as: $\mathrm{DE}_a^{\mathrm{sym}}(y|a) = \frac{1}{2}\left(\mathrm{DE}_{a_0,a_1}(y|a) - \mathrm{DE}_{a_1,a_0}(y|a)\right)$ i.e. the net treatment , which is the difference between the positive and negative effect of protected group membership. Direct discrimination exists if $\mathrm{DE}_a^{\mathrm{sym}}(y|a) > 0$ i.e. the negative effect is greater than the positive effect; **Counterfactual Indirect Effect (Ctf-IE)**: measures indirect discrimination along $A \rightarrow M \rightarrow \hat{Y}$ by holding $A$ and $C$ fixed, capturing the effect of $M$ on $\hat{Y}$. [7] define symmetric Ctf-IE as: $\mathrm{IE}_a^{\mathrm{sym}}(y|a) = \frac{1}{2}\left(\mathrm{IE}_{a_0,a_1}(y|a) - \mathrm{IE}_{a_1,a_0}(y|a)\right)$.Indirect discrimination exists if $\mathrm{IE}_a^{\mathrm{sym}}(y|a) > 0$; **Counterfactual Spurious Effect (Ctf-SE)**: measures confounding impact along $A \leftarrow C \rightarrow \hat{Y}$, varying $C$ while fixing $A$ and $M^4$. Its is given by: $SE_{a_0,a_1}(y) = P(y_{a_0}|a_1) - P(y|a_0)$. As shown in [7] **disparate treatment** (direct discrimination) exists if the symmetric difference due to $A$, $\mathrm{DE}_a^{\mathrm{sym}}(y|a)$, differs from zero. **Disparate impact** (indirect discrimination) exists if either the symmetric indirect effect, $\mathrm{IE}_a^{\mathrm{sym}}(y|a)$, or the spurious effect, $SE_{a_0,a_1}(y|a)$, is non-zero. Statistical disparity decomposes as: Mean Difference$_{a_0,a_1}(y) = \mathrm{DE}_a^{\mathrm{sym}}(y|a) + \mathrm{IE}_a^{\mathrm{sym}}(y|a) + SE_{a_0,a_1}(y|a)$. Without confounders, Ctf-DE and Ctf-IE reduce to the natural direct and indirect effects, respectively.

**Algorithmic Procedure**: [12] provide empirical formulas to estimate these effects from observed data using conditional probabilities, avoiding the need for a fully specified SCM , thus aiding ease of application (see Appendix A.4 for the empirical formulations). For each combination $m \in \mathrm{M}$ and each combination $c \in \mathrm{C}$, we get a subset of $D$ defined by $(m, c)$. For each subset $(m, c)$ : we calculate condition probability / expectation of the outcome of interest $Y_y$ for $a_0$ and $a_1$ i.e. $\mathbb{E}(y|a_0, m, c)$ and $\mathbb{E}(y|a_1, m, c)$ respectively. These are the differences in the realisation of $Y_y$ when $M$ and $C$ are the same but $A$ is different. Then for each $m$, we get the probability of $m$ given $c$ for $a_0$ and $a_1$ i.e.$P(m|a_0, c)$ and $P(m|a_1, c)$ i.e. the difference in probability of $m$ when $c$ is the same but $A$ is varied. Then lastly, for each $c$, we get it's probability for $a_0$ and $a_1$ i.e. $P(c|a_0)$ and $P(c|a_1)$ respectively. Each of these computed quantities is then combined as per (1) to (11) (see Appendix A.4) to get $\mathrm{DE}_a^{\mathrm{sym}}(y|a)$, $\mathrm{IE}_a^{\mathrm{sym}}(y|a)$, $SE_{a_0,a_1}(y|a)$ and Mean Difference$_{a_0,a_1}(y)$.

## 2.3 Counterfactual Equalised Odds (Cft-EO) : How does the protected attribute affect the model error rate?

Like counterfactual effects,[11] use the standard fairness model to define three causal counterfactual metrics based on equalized error rates: Counterfactual Direct Error Rate ($ER_{a_0,a_1}^d(\hat{y} \mid a, y) = P(\hat{y}_{a_1,y}, (\hat{P}A \setminus X)_{a_0,y} \mid a, y) - P(\hat{y}_{a_0,y} \mid a, y))$, Counterfactual Indirect Error Rate ($ER_{a_0,a_1}^i(\hat{y} \mid a, y) = P(\hat{y}_{a_0,y}, (\hat{P}A \setminus X)_{a_1,y} \mid a, y) - P(\hat{y}_{a_0,y} \mid a, y))$, and Counterfactual Spurious Error Rate ($ER_{a_0,a_1}^s(\hat{y} \mid y) = P(\hat{y}_{a_0,y} \mid a_1, y) - P(\hat{y}_{a_0,y} \mid a_0, y))$. These measure how error rates would change if the disadvantaged (advantaged) had the identity, mediators, or confounders of the advantaged (disadvantaged) group. They conclude error rates driven by $ER^d$ indicate bias, while those due to $ER^i$ or $ER^s$ are not discriminatory. [5]. Using these three counterfactual error metrics, [11] show that equalized odds can be broken down into direct, indirect and spurious components as follows: $ER_{a_0,a_1}(\hat{y} \mid y) = ER_{a_0,a_1}^d(\hat{y} \mid a_0, y) - ER_{a_1,a_0}^i(\hat{y} \mid a_0, y) - ER_{a_1,a_0}^s(\hat{y} \mid y)$.

**Algorithmic Procedure**: Unlike counterfactual effects, these metrics face a key limitation: Cft-EO cannot reliably estimate direct, indirect, and spurious effects in the presence of mediators due to identifiability issues from conditioning on both $Y$ and $\hat{Y}$ (whereas Counterfactual Effects condition only on $\hat{Y}$). The common fix is excluding $M$ from features, using only protected attributes $A$ and confounders $C$, enabling accurate estimation of $ER^d$ and $ER^s$. This solution allows for the accurate identification and estimation of Counterfactual Direct Error Rate and Counterfactual Spurious Error Rate. However, this remedial strategy is undesirable in real world applications because the exclusion of $M$ is likely to negatively impact the predictive performance of the model. Thus, to ensure that the metric is used correctly, we remove $M$ from consideration, and modify [11] to estimate the simplified counterfactual error rates as follows: $ER_{a_0,a_1}^d(\hat{y} \mid a, y) = \sum_c \left(P(\hat{y}_{a_1,c}) - P(\hat{y}_{a_0,c})\right) P(c \mid a, y)$ and $ER_{a_0,a_1}^s(\hat{y} \mid y) = \sum_c P(\hat{y}_{a_1,c}) \left(P(c \mid a_1, y) - P(c \mid a_0, y)\right)$. To compute these from the observed data $D$, we use the following procedure: For each $c \in \mathrm{C}$ we use the fitted estimator $\hat{f}$ as $\hat{f}(a_1, c)$

---

[4]Ctf-SE has no symmetric form since confounders $C$ are non-descendants of $A$ and remain unchanged under interventions.

[5]Definitions overlap with Ctf-DE, Ctf-IE, and Ctf-SE from Section 2.1, differing by focusing on error rates instead of mean difference.

and $\hat{f}(a_0, c)$ to get $P(\hat{y}_{a_1,c})$ and $P(\hat{y}_{a_0,c})$ respectively. These quantities are the differences in the realisation of $Y_y$ when $C$ is the same but $A$ is different. Then for each $c$, we get it's probability for $a_0$ and $a_1$ i.e. $P(c|a_0)$ and $P(c|a_1)$ respectively. Each of these computed quantities is then combined as per (12) to (14) (see Appendix A.5) to get $ER_{a_0,a_1}^d(\hat{y} \mid a, y)$, $ER_{a_0,a_1}^s(\hat{y} \mid y)$ and $ER_{a_0,a_1}(\hat{y} \mid y)$ .

## 2.4 Counterfactual Fairness

Unlike Counterfactual Equalised Odds and Counterfactual Effects, Counterfactual fairness[3] is an individual level causal fairness metric which is achieved if changing an individual $i$'s protected attributes doesn't change the predicted outcome $\hat{Y}_i$ i.e. $P(\hat{Y}_{A\leftarrow a}(U) = y \mid X = x, A = a) = P(\hat{Y}_{A\leftarrow a'}(U) = y \mid X = x, A = a)$

**Algorithmic Procedure**: To empirically test for counterfactual fairness first, a fully specified SCM must be defined using a specified Directed Acyclic Graph (DAG) and dataset $D$. For each instance $i$ in $D$, we retrieve the observed value $A_{\text{obs}}$ and compute a counterfactual value $A_{\text{cf}}$. We then generate samples from the SCM through two do-interventions using the standard "abduction,action,prediction"[6] procedure. First, we perform an intervention $do(A = A_{\text{obs}})$ to produce samples for the observed state $D_{do=\text{observed}}$. Second, we perform an intervention $do(A = A_{\text{counterfactual}})$ to produce samples for the counterfactual state $D_{do=\text{cf}}$. Using these samples, we fit functions $\hat{f}(D_{do=\text{obs}})$ and $\hat{f}(D_{do=\text{cf}})$ to obtain predicted outcomes $Y_{D_{do=\text{obs}}}$ and $Y_{D_{do=\text{cf}}}$. To assess counterfactual fairness, we compare the observed and counterfactual predictions. If $Y_{D_{do=\text{obs}}} \neq Y_{D_{do=\text{cf}}}$, then the prediction function $\hat{f}$ is not counterfactually fair.

**2.5 Scalability** : To address scalability, the algorithms include the following optimizations: **A. GMMs for Conditional Probability Estimation.** For Counterfactual Effects, estimating probabilities via conditional expectations on a 50,000-sample dataset takes approximately 2 seconds; using GMMs reduces this to 300–500 ms, depending on the number of features. For Counterfactual Equalized Odds, estimation takes around 1 second, with GMMs reducing latency to 300–500 ms, depending on feature count and model complexity. **B. Parallelization of Interventions.** For Counterfactual Fairness, computing for a single instance takes approximately 10 seconds without parallelization, and about 1 second with it. Actual times vary with the number of features, interventions, and the predictive model used.

# 3 Results: Application of CausalFairness to Benchmark Datasets

| Dataset | Protected Attribute | Mean Difference | FNR | FPR | $DE_a^{\text{sym}}(y\|a)$ | $IE_a^{\text{sym}}(y\|a)$ | $SE_{a_0,a_1}(y\|a)$ | $ER^d$ | $ER^i$ | $ER^s$ | Counterfactual Fairness |
|---|---|---|---|---|---|---|---|---|---|---|---|
| Adult Income | Gender | 0.203 | 0.410 | -0.104 | 0.165 | 0.039 | 0.000 | 0.000 | 0.000 | 0.000 | -0.031 |
| Adult Income | Intersectional | 0.221 | 0.445 | -0.115 | 0.152 | 0.069 | 0.000 | 0.000 | 0.000 | 0.000 | -0.068 |
| COMPAS | Race (Black) | 0.326 | -0.310 (-42) | -0.253 (-0.41) | 0.154 | 0.071 | 0.101 | FPR: -0.297, FNR: -0.265 | 0 | FPR: 0.113, FNR: 0.162 | 0.055 |
| COMPAS | Intersectional | 0.620 | -0.620 | -0.518 | 0.513 | 0.081 | 0.027 | - | - | - | 0.640 |
| LSAC | Race (Black) | 0.978 | - | - | 0.554 | 0.429 | 0.000 | - | - | - | 0.001 |
| LSAC | Intersectional | 0.990 | - | - | 0.531 | 0.458 | 0.000 | - | - | - | -0.007 |

Table 2: Statistical and Causal Fairness Metrics

**3.1 Adult Income Dataset:** We fit a logistic regression using the structure and features in Appendix A.3, Fig.1.a[6] to predict $P(\text{Income} > \$50k)$. **Counterfactual Effects:** On average, women are 20.3% less likely than men to be predicted as earning above \$50k. Most of this disparity (16.5%) is due to *disparate treatment* ($DE_a^{\text{sym}}(y \mid a)$), meaning that simply having the social identify of a woman lowers $P(\text{Income} > \$50k)$. The remaining 3.9% is due to *disparate impact* ($IE_a^{\text{sym}}(y \mid a)$) via $M$ (years of education and occupation typical of women); **Counterfactual Equalized Odds:** When refitting the model without $M$, the predictor always outputs 0, making the equalized odds and its decomposition uninformative. We thus cannot determine whether observed disparities stem from disparate treatment or impact; **Counterfactual Fairness:** As shown in Appendix Fig.2.A, the observed and counterfactual distributions do not overlap—changing a woman's gender to male shifts $\hat{Y}$ rightward, increasing $P(\text{Income} > \$50k)$. Hence, the fitted logistic regression is not counterfactually fair.

**3.2 COMPAS Recidivism Dataset:** We fit a logistic regression using the structure and features in Appendix A.3, Fig.1.b **Counterfactual Effects:** Black individuals are 32.6% more likely than white individuals to be predicted as high-risk for recidivism. Most of this is due to *disparate treatment*

---

[6]Country of residence is included as a cause of gender to replicate Zhang and Bareinboim, 2018

(15.4%), meaning that being Black alone increases $P(\text{Recidivism})$. *Disparate impact* comes from both $M$ and $C$: confounders like age and gender raise risk by $\sim 10\%$ (spurious effect), and $M$ contributes an additional 7.1%. **Counterfactual Equalized Odds:** Excluding $M$ does not make the model naive, though it increases error rates. Decomposing FPR/FNR shows most of the disparity stems from direct discrimination: 29.7% of the 41% FPR and 26.5% of the 42% FNR. **Counterfactual Fairness:** The COMPAS model is not counterfactually fair (see Appendix Fig. 2.C)—changing race from Black to white shifts the distribution of $\hat{Y}$ leftward, decreasing $P(\text{Recidivism})$.

**3.3 Law School Admission Council (LSAC) Dataset:** We fit a Random Forest regressor on GPA, LSAT, Race, and Gender to predict average grade, where $A =$(Race, Gender), $M =$(GPA, LSAT), and no $C$. **Counterfactual Effects:** The predicted average grade for the white subgroup is 0.978 higher than for the Black subgroup. Both *disparate treatment* and *disparate impact* are significant, with most of the gap (0.55) due to direct discrimination. Since there are no confounders, the direct and indirect effects correspond to the natural direct and indirect effects. **Counterfactual Equalized Odds:** Not applicable since this is a regression task. **Counterfactual Fairness:** The model is counterfactually fair, illustrating that fairness can differ at the individual vs. group level.[7]

**3.4 Intersectional Causal Fairness:** This package supports intersectional analysis to detect potential "double" or higher-order discrimination. **Adult Income:** Black women are 22.1% less likely than white men to have $P(\text{Income} > \$50k)$—2% more than the non-intersectional gender gap—with direct effect being the largest contributor. The model is also more counterfactually unfair: changing a Black woman's identity to a white man increases $P(\text{Income} > \$50k)$ by 6% (vs. 2% non-intersectionally) (see Appendix Fig. 2.B). **COMPAS:** The mean difference in $P(\text{Recidivism})$ between Black men and white women (60%) exceeds the non-intersectional racial gap, with direct discrimination as the main driver. Counterfactual unfairness also rises by $\sim 11\%$: changing a Black man's identity to a white woman lowers predicted recidivism by 64% (vs. 55%) (see Appendix Fig. 2.D). **LSAC:** The mean difference between Black women and white men is slightly higher than the non-intersectional comparison (0.99 vs. 0.978), again mainly due to direct discrimination. As in the non-intersectional case, the model remains counterfactually fair.

# 4    Limitations of CausalFairness

Generally, 1) deciding the right causal model from competing models of bias or achieving causal fairness simultaneously across multiple competing models remains an active area of research and 2) defining a hypothetical intervention on protected attributes remains a fraught process. The example application to benchmark datasets highlighted how 1) lack of identifiability can limit analysis [4] and 2) lack of methods for falsifying DAGs in the presence of competing causal models can lead to disagreements about the validity of the conclusions. For example, in the Adult Income dataset, identifiability issues prevented the causal decomposition of equalized odds. For counterfactual effects, the DAG must be Markovian; otherwise, counterfactual probabilities cannot be empirically estimated[8]. Extending the three discussed metrics to path-specific discrimination [6] is also limited by stricter identifiability constraints. Hence, causal fairness metrics should be applied cautiously,

# 5    Conclusion

This paper introduced CausalFairness - the first open source generalizable implementation for calculating key causal fairness metrics and applied it to 3 fairness benchmarking datasets. The application to benchmark datasets demonstrated how CausalFairness provides practitioners with the actionable insight - for example, at the very least the Adult Income model must eliminate at least 16.5% difference in statistical parity, while the COMPAS model needs to address 15.4% disparity in statistical parity and 29.7-26.5% in error rates (all of which can be attributed to direct discrimination), but this varies across intersectionally. Future work will expand the metrics available and extend the package to include methods for bias reduction in the causal fairness literature (see Appendix A.7).

---

[7]Our experiments also show that "fairness through unawareness"—excluding $A$ from training—can worsen fairness. For example, excluding Race in the LSAC dataset leads to a counterfactual fairness score of -0.50, meaning changing a Black individual's race to white increases the predicted average grade by 0.50.

[8]In the presence of unobserved confounding, counterfactual effects may be estimated using counterfactual randomization **?**, which is not implemented here.

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

# A Appendix

## A.1 Overview of Metrics Implemented in CausalFairness

Table 3: Overview of Metrics Implemented in CausalFairness

| Metric | Query Addressed | Level | Supported Metric Decomposition | Counterfactual Estimation Procedure |
|---|---|---|---|---|
| Counterfactual Effects (for Statistical Parity) | What would the disadvantaged (advantaged) group's acceptance rate be if they had the identity (A), mediating characteristics (M), or confounding characteristics (C) of the advantaged (disadvantaged) group? | Aggregate | Direct, Indirect, Spurious | Conditional probabilities (computed or estimated using GMM) |
| Counterfactual Equalized Odds | What would the disadvantaged (advantaged) group's error rate be if they had the identity (A), mediating characteristics (M), or confounding characteristics (C) of the advantaged (disadvantaged) group? | Aggregate | Direct, Spurious | Conditional probabilities (computed or estimated using GMM) |
| Counterfactual Fairness | What would the disadvantaged (advantaged) individual's predicted Y be if they had the identity (A), mediating characteristics (M), and confounding characteristics (C) of the advantaged (disadvantaged) group? | Individual | N/A | Predictions from functional relationships of fitted SCM |

## A.2 A Brief Literature Review

There has been considerable interest in the use of causal mechanisms to better understand black-box machine learning systems, and literature on causal fairness situates itself within the same. The causal fairness literature has three primary approaches for aiding algorithmic fairness assessment [14]: 1) Counterfactual measurement: aids the answer of what if cause-effect questions without running randomized control trials. For instance, ceritus paribus, if a woman's gender was changed to male, would her expected income be higher?; 2)Sensitivity analysis: how sensitive a model is to latent / confounding variables (which is often the status of protected attributes). For instance, sensitivity analysis can be used to "explore how sensitive our estimate of the causal link between legal representation and guilty verdict is to different levels of jury racism" and give recommendations for altering jury selection to minimize bias [14] and 3) Impact evaluation: to measure the long term consequences of automated decision making systems through the use of interventions. Following the principle of what gets measured gets managed, we recognize that causal identification of discrimination is crucial before moving on to remedial actions and impact analysis. Thus, this paper - and package - focus on addressing the gap in practical, broad adoption of causal (counterfactual) fairness metrics by providing implementations of [12]/[7], [11] and [3].

## A.3 Standard Fairness Model: Examples

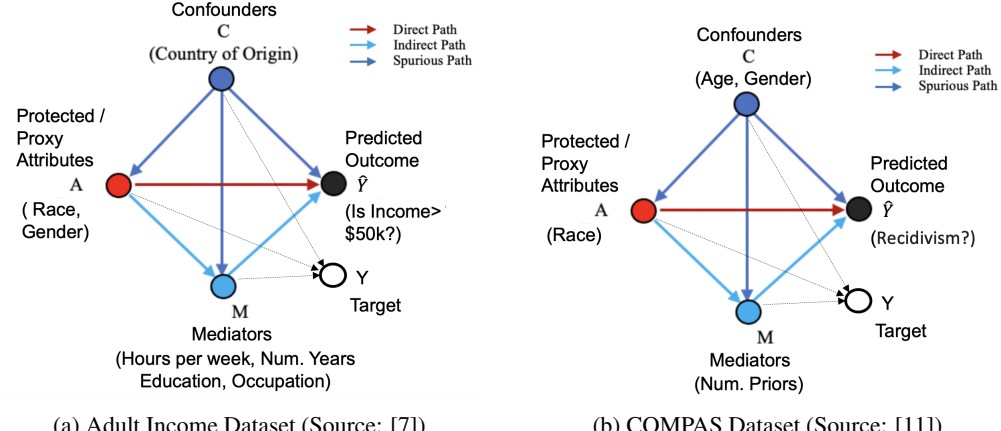

(a) Adult Income Dataset (Source: [7])    (b) COMPAS Dataset (Source: [11])

Figure 1: Standard Fairness Model

## A.4 Counterfactual effects: Empirical Formulations

Given that protected group membership can have positive and negative impacts, the direct disadvantage due to protected group membership is given by:

$$DE_{a_0,a_1}(y|a) = P(y_{a_1}, m_{a_0}, c_a|a) - P(y_{a_0}, m_{a_0}, c_a|a) \tag{1}$$

while direct advantage is given by

$$DE_{a_1,a_0}(y|a) = P(y_{a_0}, m_{a_1}, c_a|a) - P(y_{a_1}, m_{a_1}, c_a|a) \tag{2}$$

The disadvantage due to the impact of A mediated through M is given by:

$$IE_{a_0,a_1}(y|a) = P(y_{a_0}, m_{a_1}, c_a|a) - P(y_{a_0}, m_{a_0}, c_a|a) \tag{3}$$

while the advantage is given by

$$IE_{a_1,a_0}(y|a) = P(y_{a_1}, m_{a_0}, c_a|a) - P(y_{a_1}, m_{a_1}, c_a|a) \tag{4}$$

Lastly, spurious effect if given by:

$$SE_{a_0,a_1}(y) = P(y_{a_0}|a_1) - P(y|a_0) \tag{5}$$

The corresponding empirical formulations for (1) to (5) are as follows:

$$DE_{a_0,a_1}(y|a) = \sum_{c,m}(\mathbb{E}(y|a_1,m,c) - \mathbb{E}(y|a_0,m,c))P(m|a_0,c)P(c|a) \tag{6}$$

$$DE_{a_1,a_0}(y|a) = \sum_{c,m}(\mathbb{E}(y|a_0,m,c) - \mathbb{E}(y|a_1,m,c))P(m|a_1,c)P(c|a) \tag{7}$$

$$IE_{a_0,a_1}(y|a) = \sum_{c,m}\mathbb{E}(y|a_0,m,c)(P(m|a_1,c) - P(m|a_0,c))P(c|a) \tag{8}$$

$$IE_{a_1,a_0}(y|a) = \sum_{c,m}\mathbb{E}(y|a_1,m,c)(P(m|a_0,c) - P(m|a_1,c))P(c|a) \tag{9}$$

$$SE_{a_0,a_1}(y|a) = \sum_{c,m}\mathbb{E}(y|a_0,m,c)P(m|a_0,c)(P(c|a_1) - P(c|a_0)) \tag{10}$$

$$\text{Mean Difference}_{a_0,a_1}(y) = DE_a^{\text{sym}}(y|a) + IE_a^{\text{sym}}(y|a) + SE_{a_0,a_1}(y|a) \tag{11}$$

### A.5 Counterfactual Equalized Odds: Empirical Formulations

$$ER^d_{a_0,a_1}(\hat{y} \mid a, y) = \sum_c \left( P(\hat{y}_{a_1,c}) - P(\hat{y}_{a_0,c}) \right) P(c \mid a, y) \tag{12}$$

$$ER^s_{a_0,a_1}(\hat{y} \mid y) = \sum_c P(\hat{y}_{a_1,c}) \left( P(c \mid a_1, y) - P(c \mid a_0, y) \right) \tag{13}$$

Using these two counterfactual error metrics, equalized odds can be broken down into direct and spurious components:

$$ER_{a_0,a_1}(\hat{y} \mid y) = ER^d_{a_0,a_1}(\hat{y} \mid a_0, y) - ER^s_{a_1,a_0}(\hat{y} \mid y) \tag{14}$$

### A.6 Counterfactual Fairness Plots

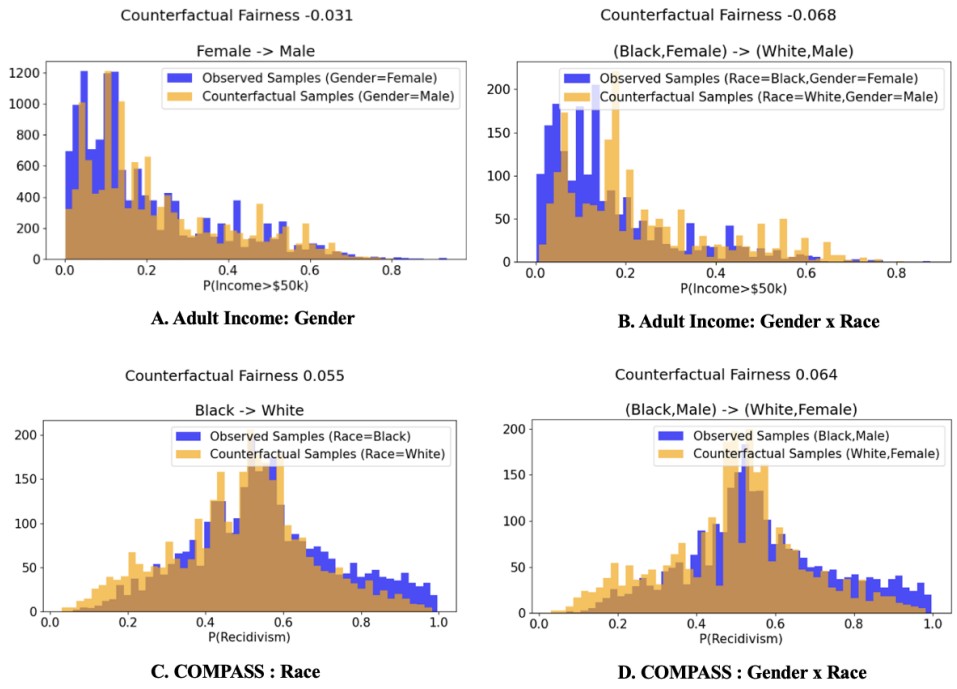

Figure 2: Counterfactual Fairness

## A.7 Bias Reduction with CausalFairness and Upcoming Future Work

But now that the causal bias has been detected, what remedial steps can practitioners take to achieve causal fairness? The easiest way to achieve causal fairness is to train an estimator using only the observable non-descendants of A [3]. However, as most observable features are likely to be descendants of A, this strategy is unsuitable. [9] Latent variables which are non-descendants of A but affect X and Y can be used to train counterfactually fair estimators [3]. Counterfactual effects [12] or counterfactual error rates [11] can be used for feature and sample selection to minimize direct, indirect and spurious discrimination. Using counterfactual fairness , a multi-world causal fairness penalty can be created to achieve counterfactual fairness under competing SCMs [8].While addressing causal bias correction algorithms is out of scope for the current paper, this is an active area of research in the causal fairness literature which we aim to incorporate into forthcoming versions of the package along side sensitivity and impact analysis.

---

[9]Linear regression which includes the protected attributes is guaranteed to be counterfactually fair [3]

