# OpenReview forum: "CausalFairness: An Open Source Python Library for Causal Fairness Analaysis"
_EurIPS.cc/2025/Workshop/UPLB — UPLB2025_

### Official Review · Reviewer_pgZW · 2025-10-27

**Rating:** 8
**Confidence:** 4

**Review:**

## Overall comment:

This paper is new software, which is a very useful contribution, and the workshop is a good venue to advertising it.

The paper is then really focused on causality and 3 measures of fairness, not on models, or learning dynamics. But this is expected for this kind of contribution.


## Soundness

This paper introduces an open source toolbox for computing fairness scores assuming a causeal structure for the data (and model) at hand. This allows to compare the fairness of different model's, based on their outputs on a given dataset, under some causality assumption.
The paper seems correct, although I did not check the details, since I am not an expert on this area (causality).

## Relevance to the workshop
This paper is in line with the following themes of the workshop:
- **Spurious correlations and shortcut learning**
- **Algorithmic bias and fairness in machine learning**

Overall, I think many authors of other papers in this conference may be interested in using this package to compare/measure fairness of their own works.

So this work contributes to making fairness a standard metric in assessing strengths of different models, which is a noble cause, and most likely in line with the workshop's goals.

## (bonus) significance

Yes, new software, which is ready-to-use, is significant, because ti can really make the difference for researchers specialized in other fields, e.g. theoreticians or ML practicionners can easily benchmark their models on the fairness metric, thanks to this (open source) software.

---

### Decision · Program_Chairs · 2025-11-03

Accept (Poster)